# Characteristics of intrinsic non-stationarity and its effect on eddy-covariance measurements of $CO_2$ fluxes

Lei Liu[1], Yu Shi[1], and Fei Hu[1,2]

[1]LAPC, Institute of Atmospheric Physics, Chinese Academy of Sciences, Beijing 100029, China
[2]University of Chinese Academy of Sciences, Beijing 100049, China

**Correspondence:** Fei Hu (hufei@mail.iap.ac.cn)

**Abstract.** Stationarity is a critical assumption in the eddy-covariance method that is widely used to calculate turbulent fluxes. Many methods have been proposed to diagnose non-stationarity attributed to external non-turbulent flows. In this paper, we focus on intrinsic non-stationarity (IN) attributed to turbulence randomness. The detrended fluctuation analysis is used to quantify IN of $CO_2$ turbulent fluxes in the downtown of Beijing. Results show that the IN is common in $CO_2$ turbulent fluxes and is a small-scale phenomenon related to the inertial sub-range turbulence. The small-scale IN of $CO_2$ turbulent fluxes can be simulated by the Ornstein-Uhlenbeck (OU) process as a first approximation. Based on the simulation results, we find that the flux-averaging time should be greater than 27 s to avoid the effects of IN. Besides, the non-stationarity diagnosis methods that do not take into account IN would possibly make a wrong diagnosis with some parameters.

## 1 Introduction

The vertical transport of carbon dioxide plays an important role in estimating the exchange of carbon dioxide between the atmosphere and other systems including the land (Horgby et al., 2019), the sea (Andersson et al., 2019), and the biosphere (Sean et al., 2021; Lauri et al., 2021). The vertical transport of carbon dioxide, dominated by turbulence mixing, can be quantified by the turbulent flux of carbon dioxide, which is normally obtained by the eddy-covariance method using high frequency wind velocity and carbon dioxide concentration measurements (Stull, 1988):

$$w'c' = (w - \langle w \rangle)(c - \langle c \rangle), \tag{1}$$

where $w'c'$ is the instantaneous turbulent flux of carbon dioxide, $w$ is the vertical wind velocity, $c$ is the carbon dioxide concentration, and $\langle w \rangle$ and $\langle c \rangle$ are the corresponding Reynolds averages. The notation $\langle \cdot \rangle$ denotes the ensemble average, i.e., averaging data collected from many independent experiments with the same conditions. It is difficult to calculate ensemble average in practice. However, if data are nearly stationary and the average time is long enough, the ensemble average can be estimated by the time average (Stull, 1988; Lenschow et al., 1994). Therefore, stationarity is a critical assumption for the eddy-covariance method and many methods are proposed to diagnose non-stationarity in the time series of instantaneous turbulent fluxes before calculating their averages (Foken and Wichura, 1996).

The non-stationarity attributed to various non-turbulent flows or external forcings has gained much attention in the literature (Mahrt and Bou-Zeid, 2020, and references therein). The non-turbulent flows or external forcings include the time changes

of surface heat fluxes (Halios and Barlow, 2018; Angevine et al., 2020), the time-dependent horizontal pressure gradients (Momen and Bou-Zeid, 2017), the sub-meso motions in the stable boundary layer (Mahrt, 2014; Sun et al., 2015; Cava et al., 2019; Stefanello et al., 2020), and so on. In fact, there is another kind of non-stationarity attributed to randomness. This kind of non-stationarity would not disappear even if the non-turbulent flows or the external forcings are absent or removed and is thus called the diffusion-like intrinsic non-stationarity or intrinsic non-stationarity (IN) (Höll et al., 2016). To our knowledge, the IN of carbon dioxide fluxes is less noticed.

In this paper, we focus on the IN of carbon dioxide turbulent fluxes in the urban boundary layer. We firstly illustrate the IN by a simple stochastic model in Sec. 2.1. Then, a method, called the detrended fluctuation analysis used to detect and quantify the IN in time series, is briefly introduced in Sec. 2.2. In Secs. 3.1 and 3.2, the IN of carbon dioxide turbulent fluxes in the urban boundary layer are analyzed and simulated. At last, we discuss the possible impacts of the IN on the calculation of carbon dioxide fluxes in Sec. 3.3.

## 2 Method and Data

### 2.1 Illustration of intrinsic non-stationarity

The intrinsic non-stationarity (IN) can be simply illustrated by the Brownian motion. A discrete time series of the Brownian motion is generated by cumulatively summing the independent Gaussian samples with zero mean and the same standard deviation $\sigma$ (Lawler, 2018):

$$B(t = N\Delta t) = \sum_{i=0}^{N} g_i,$$

(2)

where $g_i$ is a Gaussian sample and $\Delta t$ is the sampling interval. The Brownian motion $B(t)$ is non-stationary because its standard deviation scales as $\sigma\sqrt{t}$.

Two discrete time series of the Brownian motion are shown in Fig. 1a. The two series are generated by the same Brownian motion, i.e., the statistical distributions of $g_i$ are the same for the two series. However, they have different non-stationary trends: the sample A has a decreasing trend from $t \approx 10^4$ while the sample B has a wave-like trend. We call these non-stationary trends the stochastic trends because it is not attributed to any external forcings but only attributed to randomness of the time series. As a distinction, the non-stationary trends related to external forcings are called the dynamical trends. Although the stochastic trends are different, the power spectral densities $S(f)$ of two time series are not changed (see Fig. 1b): both of them agree well with the theoretical prediction that $S(f) \sim f^{-2}$ (Krapf et al., 2018). Unlike the stochastic trends, different dynamical trends indicate that systems would probably be dominated by different external forcings and the corresponding power spectral densities could also be different.

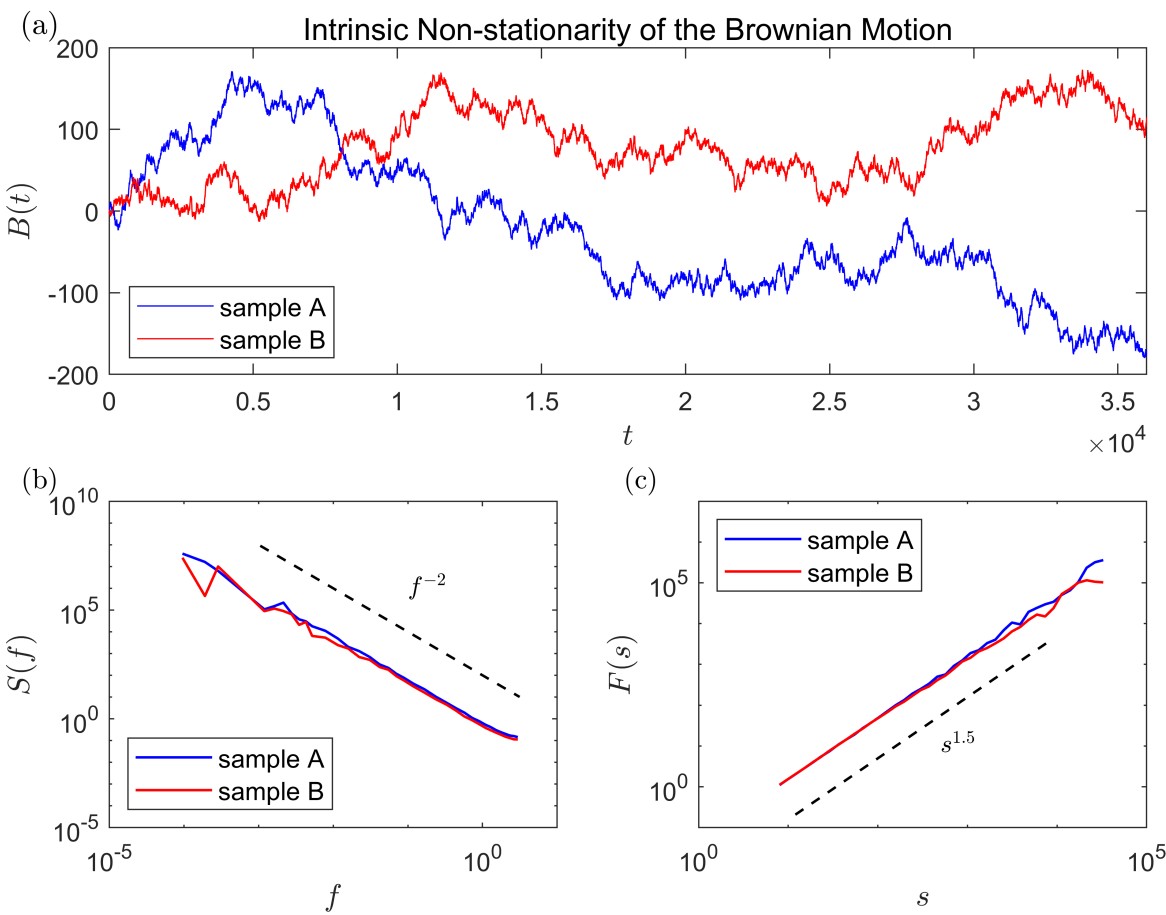

**Figure 1.** Illustration of intrinsic non-stationarity by the Brownian motion: (a) Two time series of the Brownian motion. The standard deviation of $g_i$ in Eq. (2) is set to 1. The time series length is 36000. (b) The spectra analysis of the two series, where the power spectral density $S(f)$ is plotted as a function of frequency $f$. (c) The detrended fluctuation analysis of the two series, where the fluctuation functions $F(s)$ is plotted as a function of time scale $s$. The theoretical predictions are shown by broken line in panels (b) and (c).

## 2.2 Detrended fluctuation analysis

The fluctuation analysis (FA) was firstly proposed to detect and quantify possible intrinsic non-stationarity in time series or other sequence data (Peng et al., 1992). However, the intrinsic non-stationarity and the non-stationarity caused by external forcing always coexist in a real time series. The FA can't distinguish the two kinds of non-stationarity. The detrended fluctuation analysis (DFA) method was then proposed to resolve this problem by eliminating large-scale trends in the data (Kantelhardt et al., 2001).

The DFA of a time series $x_k$ ($k = 1, 2, \cdots, N$) is briefly listed as follows. In the first step, the profile of $x_k$ is calculated by

$$Y_i = \sum_{k=1}^{i} x_k - \bar{x}, \tag{3}$$

where $\bar{x}$ is the time average of $x_k$ over the whole time period. In the second step, the profile $Y_i$ ($i = 1, 2, \cdots, N$) is cut into $N_s$ non-overlapping segments with equal time scale $s = m\Delta t$, where $\Delta t$ is the sampling interval of $x_k$ and $m$ is a positive integer ($1 \leq m \leq N$). In the third step, the profile $Y_i$ in each segment is fitted by a polynomial $p_{i,j}^n$, where $j$ is the segment index and $n$ is the degree of the polynomial. Then, the fitted polynomial in each segment is removed from the profile:

$$Y_{i,j} = Y_i - p_{i,j}^n. \tag{4}$$

In this step, the dynamical trends modeled by the polynomials are removed, but the IN stochastic trends are left (Höll et al., 2016). Generally, the choice of degree $n$ would affect the results when dynamical trends exist in the time series. However, we test the Brownian motion without dynamical trends and the carbon dioxide fluxes with dynamical trends already removed by the Reynolds average and find that the results are not substantially affected by the choice of $n$. Figure 2 shows the impact of the choice of $n$ on DFA. Results show that the choice of $n$ from 1 to 4 does not affect the conclusion of the DFA (Fig. 2a). For the Brownian motion, the fluctuation exponents are almost the same with different degrees $n$. For the carbon dioxide turbulent fluxes, the variations of the fluctuation functions also do not vary substantially with $n$ (Fig. 2b). Thus, we set $n = 1$ in this study. In the fourth step, the variance of $Y_{i,j}$ in each segment is calculated by

$$F_j^2 = \frac{1}{m} \sum_{i=1}^{m} Y_{(j-1)m+i,j}^2. \tag{5}$$

Then, the variance $F_j^2$ is averaged over all segments:

$$F(s) = \sqrt{\frac{1}{N_s} \sum_{j=1}^{N_s} F_j^2}, \tag{6}$$

where $F(s)$ is called the fluctuation function.

Generally, the fluctuation function behaves as a power function:

$$F(s) \sim s^\alpha, \tag{7}$$

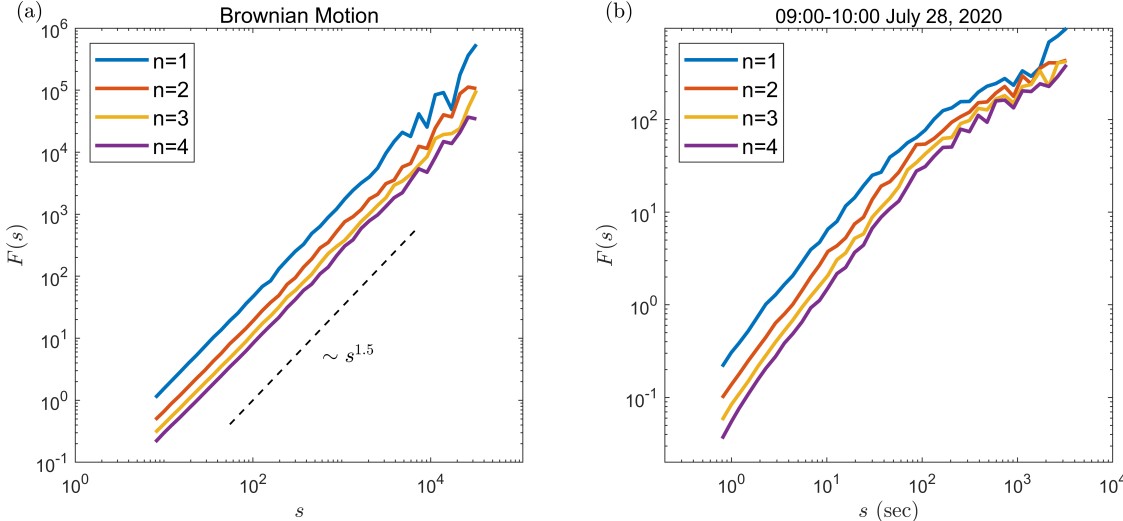

**Figure 2.** The DFA of (a) the Brownian motion and (b) the 1-hr time series of carbon dioxide turbulent fluxes with the Reynolds average time of 900 s. The results with different degrees $n$ of the polynomial in the DFA are shown by different color lines clarified in the legend. For the Brownian motion, the standard deviation of $g_i$ (see Eq. 2) is set to 1. The broken line indicates the theoretical prediction (Höll and Kantz, 2015).

where the fluctuation exponent $\alpha$ can be used to diagnose and quantify IN in the time series of $x_k$ (Kantelhardt, 2012; Løvsletten, 2017). In practice, large statistical errors will occur at large $s$. Thus, the largest fitting scale is normally set to the position

where the $F(s)$ begins to fluctuate around the power function significantly. If the fluctuation exponent $\alpha > 1$ the IN exists in the time series. The more the $\alpha$ deviates from 1, the more significant the IN is. If $1/2 < \alpha < 1$, the time series is stationary and long-term correlated. If $\alpha = 1/2$, the time series is stationary and independent (or short-term correlated). Figure 1c shows the DFA of the Brownian motion. The fluctuation exponent is close to the theoretical value of 1.5 (Höll and Kantz, 2015), which is consistent with the fact that the Brownian motion has IN. Besides, the example also shows that the IN will not be removed

in the third step of the DFA.

## 2.3 Data

The data were collected on a 325-m meteorological tower in the downtown of Beijing, China (39.97 °N, 116.37 °E). Within 5 kilometers of the tower, there are buildings with a height of about 10-60 m. About 200 m away to the west of the tower, there are a north-south highway bridge and a ring road. About 150 m away to the north of the tower, there is an east-west busy

road. The 10-Hz turbulence data, including wind velocity and carbon dioxide concentration, were collected by an ultrasonic anemometer (Windmaster Pro, Gill, UK) and an open path $CO_2/H_2O$ analyzer (LI-7500, LI-COR, USA) deployed at the 80-m level. Data collected from July 28, 2020 to August 28, 2020 are analyzed in this study.

Based on the estimation of mean building height (Oke et al., 2017), the height of the inertia sublayer around the tower is about 45 m-135 m (Cheng et al., 2018). The constant flux layer (i.e., the inertial sublayer) is observed to extent to 140 m, and the 80-m height is located in the constant flux layer (Cheng et al., 2018). According to Cheng et al. (2011), the turbulent fluctuations (with scales less than 1 min) observed on the tower are nearly isotropic, and large-scale motions (with scales greater than 1 min and less than 10 min) are anisotropic. More details about the meteorological tower, the typical meteorological conditions, urban geometry effects, and potential sources of carbon dioxide around the observation site can also be found in Cheng et al. (2018) and Liu et al. (2021).

The quality control methods proposed by Vickers and Mahrt (1997) are used to find problematic data, including spikes, dropouts, data with discontinuities, data violating absolute limits, data with the amplitude resolution problem, data with un-physical high-order moments. Their method used automated tests to identify instrumentation problems and physically plausible but unusual situations in tower time series. Besides, they also proposed automated tests to identify flux sampling problems, such as the non-stationary problem that will be discussed in the following sections. The time series seriously contaminated by the problematic data are removed in the analysis. The time series seriously contaminated by high-frequency white noises are also removed. After quality controlling, a total of 520 1-hr time series are left. The instrument reference frame is transformed to the streamline reference frame by the double rotation (Kaimal and Finnigan, 1994).

## 3 Results

### 3.1 Characteristics of intrinsic non-stationarity of carbon dioxide fluxes

The 1-hr time series of carbon dioxide turbulent fluxes is obtained by Eq. (1), where the ensemble average is replaced by the time average. In order to remove dynamical trends, the Reynolds average time is usually set to equal or be smaller than 30 minutes (Foken et al., 2004). We analyze the intrinsic non-stationarity for all the 1-hr time series of carbon dioxide turbulent fluxes, and a typical example is shown in Fig. 3. For analyzing the impact of the Reynolds average time on IN in the 1-hr time series of carbon dioxide turbulent fluxes, we here choose the Reynolds average times $\tau = 900$ s, and 300 s that are commonly used in the eddy-covariance method (Doran, 2004; Metzger et al., 2007; Li and Bou-Zeid, 2011; Donateo et al., 2017). In order to show the effect of very small Reynolds average time in sharp contrast, we also choose a timescale of 6s in the analysis.

The DFA is shown in Fig. 3b. Two scaling regimes are found in the fluctuation functions. At large time scale $s$, the fluctuation exponent is found to be less than 1; at small time scale $s$, the fluctuation exponent is found to be greater than 1. Results indicate that the time series of carbon dioxide turbulent fluxes have IN at small time scales but are stationary at large time scales, whatever the Reynolds average time is. As shown in Fig. 3a, the small-scale variations of these time series are evidently non-stationary, although the large-scale dynamical trends have been removed by subtracting the Reynolds average from data. Besides, one can note that the fluctuation functions with $\tau = 900$ s and 300 s are almost the same but are different from that with $\tau = 6$ s. The crossover scale in the case with $\tau = 6$ s (at $s \approx 2$ s) is smaller than that in cases with $\tau = 900$ s and 300 s (at $s \approx 20$ s). The power spectral densities of these time series are shown in Fig. 3c. The spectra with $\tau = 900$ s and 300 s are almost the same but are also different from that with $\tau = 6$ s. The case with $\tau = 6$ s is found to have a much shorter inertial sub-range

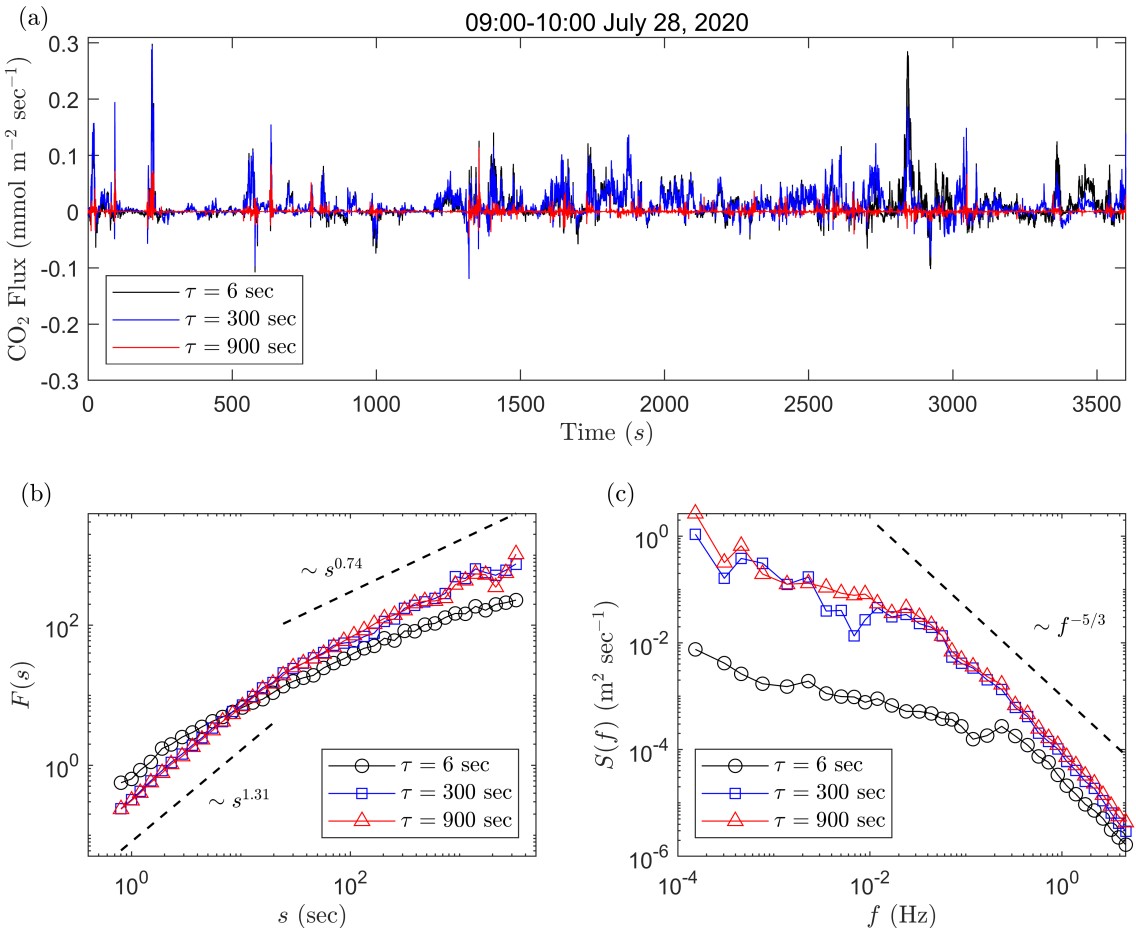

**Figure 3.** The intrinsic non-stationarity in the 1-hr time series of carbon dioxide turbulent fluxes: (a) The 1-hr time series of instantaneous turbulent fluxes of carbon dioxide with the Reynolds average time $\tau$=900 s, 300 s, and 6 s. (b) The detrended fluctuation analysis of these time series. For comparison, the time series are normalized to zero mean and unit variance. The power functions with the fitted fluctuation exponents are shown by the broken lines. (c) The power spectral densities of these time series. The Kolmogorov $-5/3$ law is shown by the broken line.

than cases with $\tau = 900$ s and 300 s. The inertial sub-range is recognized by the Kolmogorov $-5/3$ law (Kolmogorov, 1941). Results indicate that the IN is a small-scale phenomenon which is intimately related to the inertial sub-range turbulence. The choice of very small Reynolds average time could partly remove the IN, but the turbulence contribution to fluxes is also partly removed. It is believed that if the sampling frequency is improved and the flux-averaging time is further reduced, the stationary assumption of the eddy-covariance method can be better guaranteed. Our findings indicate that the above consideration may not be right because the further reduction of flux-averaging time would face the intrinsic non-stationarity.

### 3.2 Simulation of intrinsic non-stationarity

The Ornstein-Uhlenbeck (OU) process, that is well-studied and used to model many physical and chemical processes (Gardiner, 1985), is a simple model of small-scale intrinsic non-stationarity (IN). The OU process has similar crossover characteristic as carbon dioxide fluxes. Besides, many statistical properties (including the fluctuation exponents) of the OU process can be solved analytically (Czechowski and Telesca, 2016). We here use this model to simulate the IN of carbon dioxide fluxes.

The discrete time series of the OU process is generated by the iterative equation:

$$y(t + \Delta t) = y(t) - ay(t)\Delta t + b\sqrt{\Delta t}\xi, \tag{8}$$

where $a$ and $b$ are model parameters, $\Delta t$ is the sampling interval, and $\xi$ is an independent random variable with the normal distribution. For the OU process, the fluctuation function $F(s) \sim s^{0.5}$ at large scales and $F(s) \sim s^{1.5}$ at small scales (Höll and Kantz, 2015; Czechowski and Telesca, 2016; Løvsletten, 2017). This indicates that the OU process has IN at small scales but is stationary at large scales, as clearly illustrated by an example in Figs. 4 and 5. Figure 4a shows the 1-hr time series of the OU process generated by Eq. (8). Due to the small-scale IN, the time series seems to be intermittent. However, the large-scale variations of the same OU process, obtained by averaging the time series in Fig. 4a with an average time much greater than the crossover scale, seem to be like a stationary white noise (Fig. 4b). The DFA shows that the fluctuation exponent of the averaged time series is about 0.5, as the fluctuation exponent of the unaveraged time series at large scales (Fig. 5). This indicates that the averaged time series with a large average time, reflecting the large-scale variations of the OU process, is stationary.

The DFA of 520 1-hr time series of instantaneous carbon dioxide fluxes is shown in Fig. 6. The Reynolds average time is set to 5 min. Results show that the fluctuation functions of carbon dioxide turbulent fluxes typically have two scaling regimes, as shown in Fig. 3. The fluctuation exponents are generally greater than 1 at small scales and less than 1 at large scales. The OU process can fit the data as a first approximation, although the fluctuation exponent of data seems to be greater at large scales and less at small scales, compared with the OU process. The details of fitting procedure are listed as follows. In the first step, choose the parameters of the OU process $a$ and $b$ from the same set $(0.1, 0.2, \cdots, 1)$, and set $\Delta t = 0.1$ s. The 1-hr time series of the OU process is generated by Eq. (8) with the chosen parameters. In the second step, compute the fluctuation function of the generated 1-hr time series. In the third step, go back to the first step and choose another new value of $a$ or $b$ in the set $(0.1, 0.2, \cdots, 1)$. If all possible combinations of $a$ and $b$ are used, go to the fourth step. In the fourth step, the root mean relative

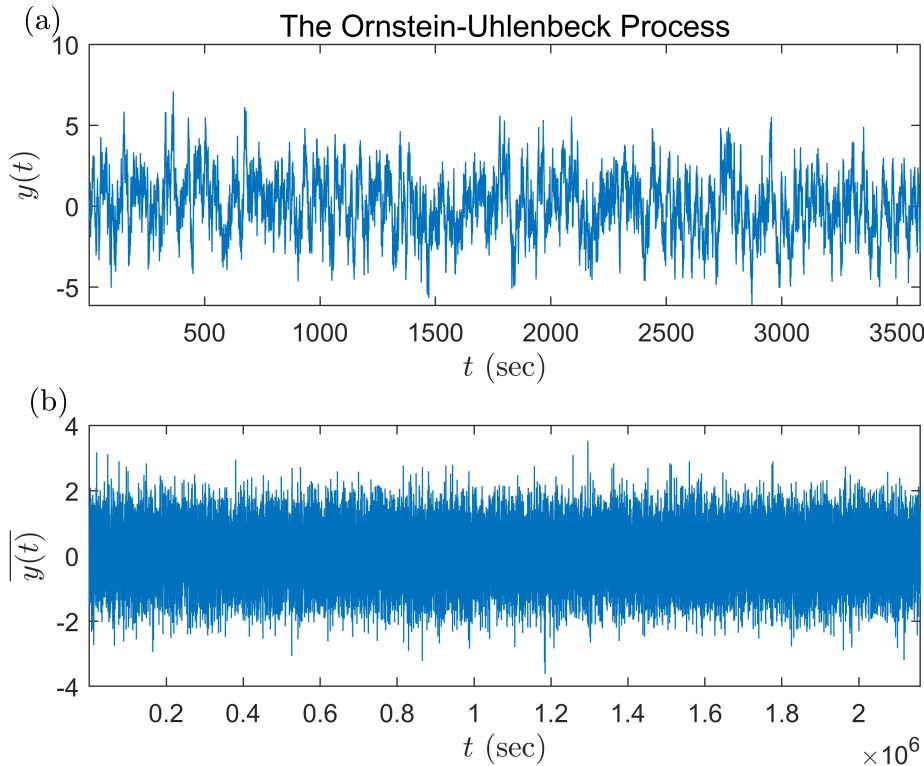

**Figure 4.** Small-scale non-stationarity and large-scale stationarity in the same OU process. (a) The 1-hr time series of the OU process with $a = 0.15$, $b = 1$, and $\Delta t = 0.1$ s. (b) The average time series of the OU process with the same parameters as in (a). The average time is set to 1 min.

square error for the $i$-th combination of a and b is computed:

$$RMRS_i = \sqrt{\frac{1}{N_j} \sum_{j=1}^{N_j} \left[ \frac{F_i(s_j) - F_{data}(s_j)}{F_i(s_j)} \right]^2}, \tag{9}$$

where $N_j$ is the number of discrete time scale $s_j$, $F_i$ is the fluctuation function of the OU process with the $i$-th combination of $a$ and $b$, and $F_{data}$ is the averaged fluctuation function of carbon dioxide turbulent fluxes (shown by the red line in Fig. 6). In the fifth step, the parameters of $a$ and $b$ corresponding to the minimum of $RMRS_i$ are considered as the optimal fitting parameters.

The fluctuation exponent of the OU process at large scales equals to 0.5. The fact that the fluctuation exponent of data is greater than that of the OU process but less than 1 at large scales indicates that the data is stationary and long-term correlated at large scales. This could be related to the large-scale coherent structure of scalar turbulence (Celani and Seminara, 2005; Liu and Hu, 2020). The fluctuation exponent of the OU process at small scales equals to that of the Brownian motion. The fluctuation exponent of data seems to be less than that of the OU process at small scales which indicate that the data deviate

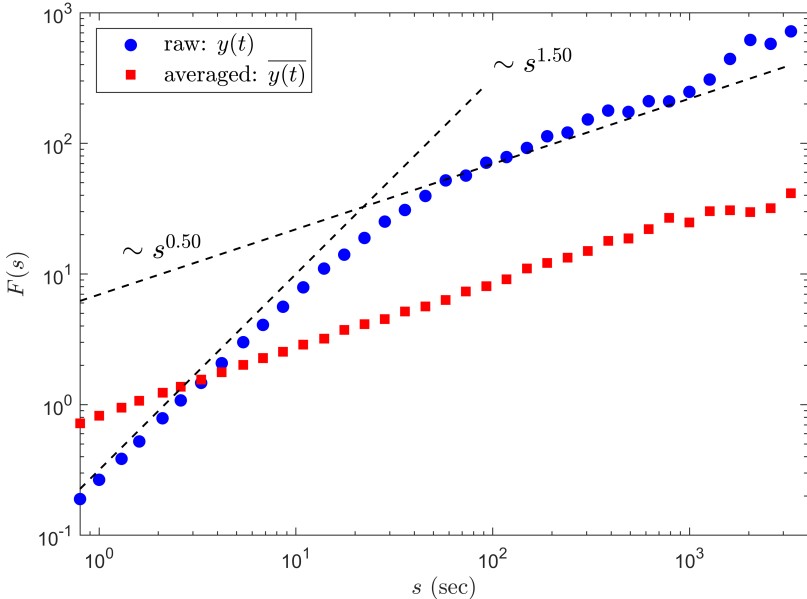

**Figure 5.** The DFA of the time series shown in Fig. 4. The fluctuation functions of the unaveraged and averaged time series are denoted by blue circles and red rectangles respectively. The theoretical predictions of the OU process are also denoted by broken lines (Höll and Kantz, 2015; Czechowski and Telesca, 2016; Løvsletten, 2017).

from the Brownian motion at small scales. This could be related to the non-Gaussian intermittency of turbulence in the inertial subrange (Liu et al., 2019). As we have discussed in Sec. 3.1, the IN is intimately related to the inertial sub-range turbulence which is usually considered to be produced by the cascade mechanism (Kolmogorov, 1941). The OU process is a very simple
mathematical model that does not include the cascade mechanism. It is believed that the fitting results would be improved by adding the cascade mechanism into the OU process. This paper focuses on the main characteristics of the IN, and further extensions of the OU process will be investigated in a future study.

### 3.3 Impacts of intrinsic non-stationarity on flux calculation

There are at least two impacts of intrinsic non-stationarity (IN) on the calculation of average carbon dioxide turbulent fluxes.
First, the IN could affect the short-term averaged turbulent flux normally used in the analysis of plant photosynthesis efficiency (Van Kesteren et al., 2013). To avoid IN at small scales, the average time averaging instantaneous turbulent fluxes (i.e., the flux-averaging time) should be much greater than the crossover scale in the fluctuation function, because crossover scale separates the IN at small scales and stationarity at large scales. Note that the flux-averaging time is not necessarily the same as the Reynolds average time (Foken et al., 2004). The former is denoted by $T$ in the following discussion. For the
Ornstein-Uhlenbeck (OU) process (Czechowski and Telesca, 2016), the crossover scale

$$s_{\times} \approx \frac{5.4}{a}. \tag{10}$$

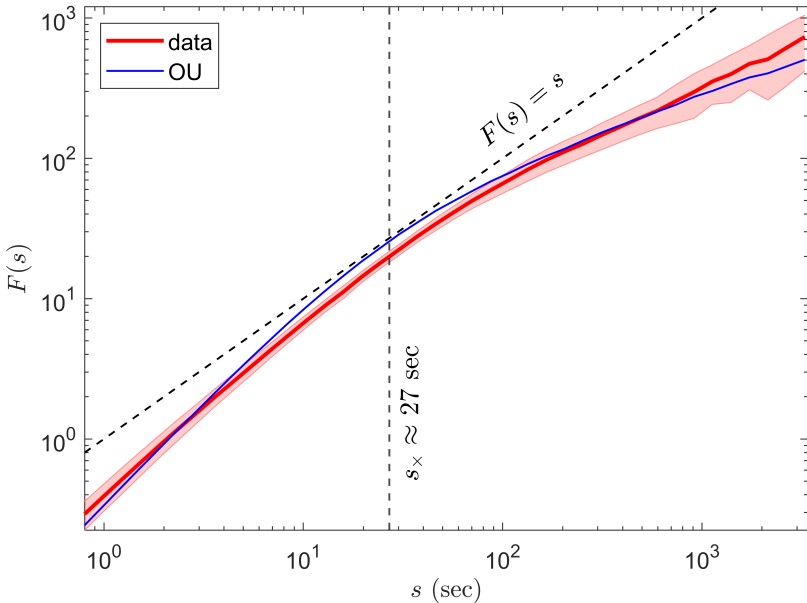

**Figure 6.** The detrended fluctuation analysis of all 1-hr time series of carbon dioxide fluxes. The Reynolds average time is set to 5 min to calculate fluxes. The sample-averaged fluctuation function is shown by the red line and uncertainties estimated by the standard deviation are shown by the red shading. The fitted fluctuation function of the Ornstein-Uhlenbeck (OU) process is shown by the blue line. The fitted parameters $a = 0.2$, $b = 0.7$, and $\Delta t = 0.1$ s. The vertical broken line indicates the crossover scale estimated by Eq. (10). For comparison, the function of $F(s) = s$ is also shown by the broken line.

According to the fitting results in Fig. 6, the crossover scale of carbon dioxide turbulent fluxes is about 27 s. The errors of fluxes averaged with $T \lesssim s_\times \approx 27$ s would be large due to the existing of small-scale IN.

Second, the IN could affect the diagnosis methods of non-stationarity. For example, Vickers and Mahrt (1997) have used a dimensionless index $RN$ to diagnose non-stationarity:

$$RN = \frac{\delta x}{\bar{x}}, \tag{11}$$

where $\delta x$ is the difference between the beginning and the end of linear regression trend of the diagnosed time series and $\bar{x}$ is the time average of the same time series. If $RN$ is greater than a predefined threshold, the time series is diagnosed as non-stationary and are not recommended to be averaged by time. We here use the $RN$-method to the OU process. Because the OU process is stationary at large scales, it is meaningful to calculate its average with a large average time. Thus, we hope that the OU process can be diagnosed as stationary by the $RN$-method. The proportion of diagnosed stationarity for the OU process is

plotted as a function of threshold in Fig. 7. Results show that once the threshold is less than a critical value the $RN$-method has a certain probability to make wrong diagnosis. With the decrease of the threshold, the probability of misdiagnosis will increase. The critical threshold increases as the parameter $a$ decreases. In the limit case with $a = 0$, the OU process with small-scale IN becomes the Brownian motion with full-scale IN (see Eq. 8). We thus hope that the proportion of diagnosed stationarity for the

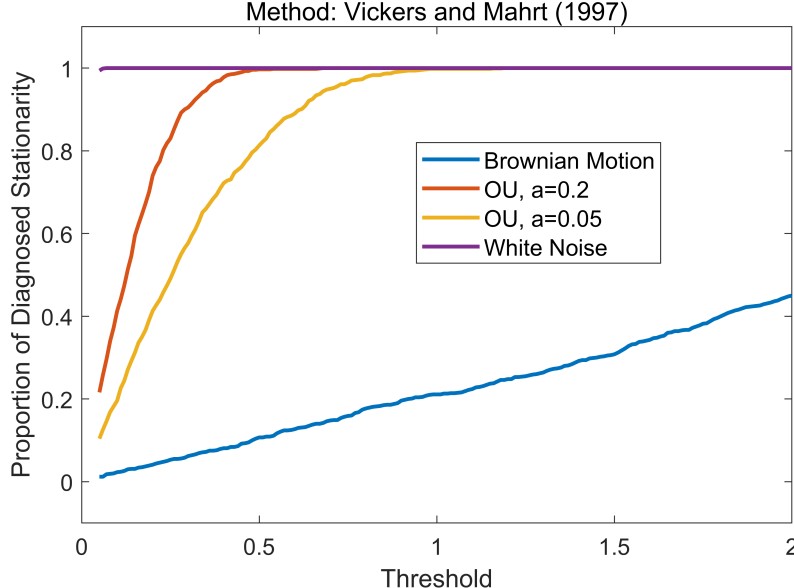

**Figure 7.** The impact of IN on the non-stationarity diagnosis method proposed by Vickers and Mahrt (1997). The proportion of diagnosed stationarity is plotted as a function of threshold. The functions for the white noise, the OU process with $a = 0.2$ and $b = 0.7$, the OU process with $a = 0.05$ and $b = 0.7$, and the Brownian motion are shown by different color lines, as listed in the legend. The number of generated time series of each model is 1000. To avoid zero denominator in Eq. (11), the averages of all generated time series are set to 1.

Brownian motion is 0; however, the $RN$-method has the probability of misdiagnosis almost at any threshold. In another limit case of the white noises without non-stationarity, the $RN$-method performs well and the probability of misdiagnosis is 0 for most thresholds. The results remind us that the parameters of diagnosis methods must be carefully chosen when diagnosing carbon dioxide fluxes with IN.

## 4   Conclusions

We analyze the time series carbon dioxide fluxes observed by the eddy-covariance system in the downtown of Beijing and find a new kind of non-stationarity less discussed in the literature. As illustrated by the Brownian motion, the new kind of non-stationarity has nothing to do with non-stationarity attributed to non-turbulent flows or external forcings; therefore, it is called the intrinsic non-stationarity (IN). The detrended fluctuation analysis (DFA) is a useful method to measure IN in real time series where IN always coexists with non-stationarity by external forcings. The DFA shows that the instantaneous turbulent fluxes of carbon dioxide have IN at small time scales. Combined with the spectral analysis, the IN is found to be related to inertial sub-range turbulence. The small-scale IN can be simulated by the Ornstein-Uhlenbeck (OU) process as a first approximation. The potential impacts of IN on the calculation of turbulent fluxes are also discussed. According to the OU process, the crossover scale, that is the characteristic scale under which the IN cannot be ignored, is estimated to be about 27 s. Thus, the IN could

contribute systematical errors to short-term averaged fluxes when the average time is not much greater than the crossover time. Besides, we also find that there may be a probability of misdiagnosis when applying some non-stationarity diagnosis method

to the time series with IN. Thus, IN should be seriously considered when designing new diagnosis methods.

This work only focuses on the main characteristics of IN of carbon dioxide fluxes in the urban boundary layer. It is interesting to discuss the difference characteristics of IN between the urban and rural boundary layer. The relationships between the IN characteristics (e.g., the crossover scale and fluctuation exponents) and urban boundary layer parameters (e.g., stability, roughness, boundary-layer height) should be systematically studied. The extensions of the OU process should be tried to

210 obtain a better fitting with data. Except for the carbon dioxide turbulent flux, is there IN in other turbulent fluxes with different terrains? The above problems remain to be resolved in the future study.

*Code and data availability.* The Matlab code of the DFA is provided by Martin Magris (downloadable at https://www.mathworks.com/matlabcentral/fileexchange/67889-detrended-fluctuation-analysis-dfa). A total of 520 1-hr time series of carbon dioxide turbulent fluxes used in this study are available online at https://doi.org/10.4121/14790084.v1

*Author contributions.* FH and LL conceived the idea; LL finished all analysis and wrote the manuscript; YS contributed to revise the manuscript and edit the plots. All authors contributed to the interpretation of the results.

*Competing interests.* The authors declare that they have no conflict of interest.

*Acknowledgements.* This work was supported by the National Natural Science Foundation of China (Grant NOs. 42175101 and 41975018) and the General Financial Grant from the China Postdoctoral Science Foundation (2020M670420).

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
