# Peer review of "Characteristics of intrinsic non-stationarity and its effect on eddy-covariance measurements of CO2 fluxes"

_Nonlinear Processes in Geophysics, 2021_

## Author Comment (AC2)

1) One major concern about the analysis: Lines 62-75: As far as I can tell, the definition of the segments of length "s" implies that not the same data is used for calculating F(s) for each "s" as "N" is not a multiple for all "s". How does this affect the results, in particular, for larger s (for instance, s> N/2)?

**Response:** As the referee commented, the amount of data used for calculating $F(s)$ for each 's' is different. This is a common problem that many time-series analysis methods encounter. Generally, as long as the scale 's' is not particularly large, it is commonly considered that results would not be seriously affected by the amount of data. The same is true for the DFA. According to numerical simulations of many mathematical models, Kantelhardt et al. (2002) recommend that $s < N/4$, because for large 's' statistical errors become large. In practice, one can approximately define the maximum of 's' for fitting by the position where $F(s)$ fluctuates significantly around the power law. From Fig. 2 in our paper, one can see that the large fluctuations of $F(s)$ appear at $s \approx 1000$ sec. Thus, the maximum of 's' for fitting is set to be 1000 sec in our analysis. Because the crossover scale is about 20-30 sec that is far less than the maximum of 's' for fitting, the statistical errors caused by large 's' would not affect the results in this study.

In order to facilitate readers to understand the analysis method of this paper, above discussions have been briefly added in lines 75-76 in the revised paper.

Kantelhardt JW, Zschiegner SA, Koscielny-Bunde E et al. (2002), Multifractal detrended fluctuation analysis of nonstationary time series. Physica A, 316: 87–114.

2) Another concern is that, although being a reasonable approximation to the observations, the simple OU process could be fit better through some extensions (as mentioned in lines 144-145). Why are they not tested in this study? At the very least the physical reasons for the difference between the observations and the OU should be discussed? These physical insights should inform the extensions the OU requires in order to fit the data better.

**Response:** There may be some misunderstandings about our statements in lines 144-145. We meant to say that the OU process can be used as a starting point for future study. Based on our analysis of the OU process, one would find a certain kind of extension to fit the data better than the OU process. However, until now we still don't know this kind of extension. To avoid misunderstanding, we have deleted these statements in the revised paper.

Our paper indicates that the intrinsic non-stationarity (IN) is intimately related to the inertial-subrange turbulence which is normally considered to be generated by the cascade process. After all, the OU process is just a simple mathematical model and does not consider the cascade process. Thus, we also believe that the extension could be found by adding the cascade process into the OU process. This paper only focusses on the main characteristics of IN. The complicated physical mechanism of IN and the corresponding improved fittings would be reported in future work.

Above discussions have been briefly added in lines 160-164 in the revised paper.

3) Line 7: "... we find that the average time should be..." - Do you mean the average time for eddy covariance measurements?

**Response:** Yes, it is the averaging time for eddy covariance measurements. To be precise, it is the averaging time used to average the instantaneous turbulent flux and is usually called the flux-averaging time in literature. We have clearly interpreted this term in Sec. 3.3. For clarity, we have revised 'the average time' to 'the flux-averaging time'.

4) Lines 11-18: The first paragraph and the first two lines of the second paragraph are not related to the rest of the paper. The paper deals with quantifying the intrinsic non-stationarity of carbon fluxes. Such quantification is not related to climate change nor does a better understanding of it allow for reductions in anthropogenic greenhouse

gas emissions. In particular, lines 17-18 are unclear how a better understanding of the CO2 transport is helpful in reducing the CO2 reductions. I guess the authors refer to air quality and not reduction in CO2 emissions.

**Response:** As the reviewer commented, the first paragraph is less related to the rest of the paper. Thus, we decided to delete the first paragraph in the revised paper.

The first two lines of second paragraph are intended to briefly explain the significance of the research subject of this paper, that is, the CO2 vertical exchange. However, the sentences are too ambiguous to cause misunderstanding. We have rewritten these sentences in the revised paper.

5) Lines 93-94: "The time series seriously contaminated by high-frequency white noise are also removed?" Why? How does white noise affect the analysis?

**Response:** We remove the high-frequency white noises because they are not real signals of wind speeds and carbon dioxide concentrations. At high frequency, both the spectra of wind speeds and carbon dioxide concentrations should obey a power law with the power exponent of -5/3. However, the spectrum of time series seriously contaminated by high-frequency white noises will be flat at high-frequency range. The white noises may be caused by the random electrical signal noise of anemometer or the mechanical resonance or other reasons. In this paper, we do not pay special attention to the effect of noise and any other kind of problematic data. We just discard all of them in the analysis, as usually done in calculations of turbulent fluxes.

6) Line 103: Why do you choose these Reynolds averaging timescales? In particular, 6 seconds is not a standard value to decompose the turbulent fluxes from the mean flow.

Line 110: "The crossover scale in the case with 6 seconds is smaller than that in cases with 900 and 300 seconds." - what does this tell us? What is the physical interpretation?

Lines 114-115: "Results indicate that the IN is a small-scale phenomenon which is intimately related to the inertial sub-range turbulence" - Isn't that expected?

**Response:** The Reynolds averaging timescales of 900s (15 min) and 300s (5min) are commonly used in the eddy-covariance method (for example, see Doran, 2004; Metzger et al, 2007; Li and Bou-Zeid, 2011; Donateo et al, 2017). The Reynold averaging filters out scales greater than the averaging timescales. The intrinsic non-stationarity (IN) is related to scales (crossover scale) much smaller than the averaging timescales. Thus, the fluctuation functions do not significantly change with large Reynold averaging timescales, as shown in Fig 3.

We note that the crossover scale is about 20-30 sec. Thus, if we use a Reynolds averaging timescale much smaller than the crossover scale, the crossover phenomenon would be changed. In order to show this phenomenon in sharp contrast, we choose a timescale of 6s far away from the crossover timescale. We indeed observed that the crossover moves to a smaller scale. Someone believes that if the sampling frequency is improved and the flux-averaging timescale is further reduced, the stationarity assumption of the eddy-covariance method can be better guaranteed. Our findings indicate that the above consideration may not be right because the further reduction of flux-averaging timescale would face another new kind of non-stationarity (that is IN) and would partly remove the contribution of turbulence.

Above discussion have been briefly added in the end of Sec. 3.1 in the revised paper.

The statement of L114-115 is indeed an expected conclusion. However, we still need to do spectral analysis to confirm this conclusion (see Fig. 3c). The significance of this conclusion is to tell us that the IN is not rooted in large-scale meteorological processes, but rooted in the nonlinear and non-Gaussian cascade process of inertial-subrange turbulence. It is the inherent property of turbulence. As we have said in the second respond, this conclusion gives us a possible direction to improve the OU process.

Doran, J.C. (2004). Characteristics of Intermittent Turbulent Temperature Fluxes in Stable Conditions. Boundary-Layer Meteorology 112, 241–255.

Metzger M, McKeon B.J and Holmes H (2007) The near-neutral atmospheric surface layer: turbulence and non-stationarity. Phil. Trans. R. Soc. A.365, 859–876.

Li, D., Bou-Zeid, E. (2011). Coherent Structures and the Dissimilarity of Turbulent Transport of Momentum and Scalars in the Unstable Atmospheric Surface Layer. Boundary-Layer Meteorol 140, 243–262.

Donateo, A., Cava, D. and Contini, D (2017). A Case Study of the Performance of Different Detrending Methods in Turbulent-Flux Estimation. Boundary-Layer Meteorol 164, 19–37.

7) Lines133-134: "..., although the fluctuation exponent of data seems to be greater at large scales and less at small scales, compared to the OU process." What does this tell us? What are the physical reasons for that behaviour?

**Respond**: As we have stated in the end of Sec. 2.2, if the fluctuation exponent $0.5 < \alpha < 1$, the time series is stationary and long-term correlated. If the fluctuation exponent $\alpha = 0.5$, the time series is stationary and independent.

The fluctuation exponent of the OU process at large scales equals to 0.5. The fluctuation exponent of data seems to be greater than that of the OU process but less than 1 at large scales, which indicates that the data is stationary and long-term correlated at large scales. This could be related to the large-scale coherent structure of scalar turbulence (Celani and Seminara, 2005; Liu and Hu, 2020).

At small scales, the OU process approaches the Brownian motion, that can be easily inferred from Eq. 8 in the paper:

$$\lim_{\Delta t \to 0} \Delta y = b\sqrt{\Delta t}\xi,$$

where $\xi$ is an independent random variable with the normal distribution. Thus, the fluctuation exponent of the OU process at small scales equals to that of the Brownian motion. The fluctuation exponent of data seems to be less than that of the

OU process at small scales, which indicate that the data deviate from the Brownian motion at small scales. This could be related to the ubiquitous non-Gaussian intermittency of turbulence at the inertial subrange (Liu et al., 2019).

Above discussion have been added in the end of Sec. 3.2 in the revised paper.

Liu L., Hu F., Huang SX (2019). A Multifractal Random-walk Description of Atmospheric Turbulence: Small-scale Multiscaling, Long-tail distribution, and Intermittency. Bound.-Layer Meteor., 172, 351–370,

Liu L., Hu F. (2020). Finescale Clusterization Intermittency of Turbulence in the Atmospheric Boundary Layer, Journal of the Atmospheric Sciences, 77: 2375-2392.

Celani A. and Seminara A. (2005). Large-scale Structure of Passive Scalar. Phys. Rev. Lett., 94:214503

8) Line 4: "widespread" sounds as if a spatial analysis has been done which is not the case. I suggest using "common".

**Response:** We have replaced the word 'widespread' by 'common' in the revised paper. Thank you.

9) Correct the citation style. At most places the references should be (Author, year), e.g. line 15

**Response:** That is caused by a typesetting software error. We have checked the entire document carefully and corrected all errors in the revised paper. Thank you.

10) Lines 63: "m is a positive integer" - I guess 2<= m <= N? This goes back to general comment about the accuracy of estimating F(s) as s gets large

**Response:** Yes, m is a positive integer and 2<= m <= N. In practice, we choose a value of m much smaller than N to fit the $F(s)$, because the statistical errors get large at large values of m. The maximum of m for fitting is changed case by case.

We normally plot $F(s)$ with m from 2 to N. The maximum of m can be defined by the position where $F(s)$ begins to fluctuate around the power law significantly.

11) Line 84: "The data were…" suggestion: "Carbon dioxide turbulent fluxes were …"

**Response:** The 'data' are referred to the raw data including wind velocity and carbon dioxide concentrations. Due to this reason, we intend to use 'data'.

12) Lines 91-93: Hard to understand the 2 sentence: "The quality control methods, proposed by Vickers and Mahrt and including spikes…" Do you mean: "The quality control methods proposed by Vickers and Mahrt (1997) are applied to remove problematic data with spikes, dropouts, ..."?

**Response:** We have revised the statement as the referee suggested.

13) Figure 3: coordinate the colors between the different subplots such that 6 sec, 300 sec, or 900 sec have the same color in each subplots

**Response:** We have coordinated the colors between the different subplots in the revised paper.

14) Line 115: "The choice of very small Reynolds …" - word missing (averaging timescales?)

**Response:** Yes, some words are missed here. We have corrected this mistake in the revised paper.

---

## Author Comment (AC3)

1) In general the discussion remains mostly on a technical leveland not much interpretation of the physical processes related to the intrinsic nonstationarity is offered. Also, while data collected in an urban boundary layer are used, specific considerations of the effect of the urban complexity are missing in the analysis. This could be possibly done by proposing the analysis also for data collected in a homogeneous site, when available: the authors might consider performing this additional work.

**Response:** The reviewer has proposed many important research directions. In this paper, we have demonstrated that the intrinsic non-stationarity (IN) is a small-scale inertial sub-range phenomenon and would be an intrinsic property of turbulence. The physical process of IN may be related to the non-Gaussian cascade process of small-scale turbulence. The understanding of this process relies on the in-depth and systematic knowledge of basic characteristics of IN.

This paper only focuses on the main characteristics of IN in the CO2 turbulent fluxes and its effect on the calculation of CO2 turbulent fluxes, due to the limited data collected in this work (only single-layer turbulent wind speed and CO2 concentrations are collected). The purpose of this paper is to arouse the attention of readers to the systematic errors of the eddy-covariance method caused by IN. About the physical process of the IN, the influence of the urban boundary layer on the IN, and the characteristics of IN on different terrains, we hope to discuss these problems in other papers after collecting more data and running numerical simulations.

Considering the importance of the problems proposed by the reviewer, we added them in the form of outlook in the Conclusion of the revised paper. In order to clarify the subject of this paper, we revised the title as 'Characteristics of intrinsic non-stationarity and its effect on eddy-covariance measurements of CO2 fluxes'.

2) L11-16: the first part of the introduction is rather general and goes far beyond the specific topic and purpose of the work presented: a more focused introduction would be worth, with more specific references to the related literature.

**Response:** As the reviewer commented, the first paragraph goes far beyond the specific topic and purpose of the work. Thus, we decide to delete the first paragraph in the revised paper. The first two lines of the second paragraph are intended to briefly explain the significance of the research subject of this paper, that is, the $CO_2$ vertical exchange. However, the sentences are too ambiguous to cause misunderstanding. We have rewritten these sentences and add the related literature in the revised paper.

3) L58-60: the improvement achieved using the DFA method and the reasons to choose it should be better introduced and explained, citing the original publication is not enough.

**Response:** The statements are indeed somewhat inaccurate and vague. The fluctuation analysis (FA) was firstly proposed to detect possible non-stationarity in the data. However, the intrinsic non-stationarity and the non-stationarity caused by external forcing always coexist in a real time series. The FA can't distinguish the two kinds of non-stationarity. The DFA method was then proposed to resolve this problem by detrending large-scale trends in the data. That's why we choose this method in this analysis. We have corrected the misleading statements in the revised paper.

4) L72: is there any specific reason why n=1 was chosen for this study?

**Response:** The n=1 was chosen because it uses the simplest interpolation (linear interpolation) and has the similar results as high-order interpolations (see Fig. 3a).

5) L84-90: besides the description of the instrumentation, given that the data cover a period of just one month in Summer time, some details on the typical meteorological conditions of the area and on how the urban geometries may interfere with and affect the incoming flow would be worth. Also, being Summer time, one can expect that the main pollutant emissions are due to traffic and possible

industries: some information on this could be added.

**Response:** In fact, the topics mentioned by the reviewer, such as the typical meteorological conditions of the observation site, the effect of the urban geometries and the carbon emissions of the observation, have been extensively studied in Cheng et al (2018) that has been cited in the paper. Cheng et al (2018) used about 4 years and multi-layer data in their analysis. We only collected one month single-layer data and some data have been discarded by the quality control. Besides, as we have mentioned in the first response, the research subject of this paper does not involve the impact of meteorological factors and emission sources on IN. Therefore, we intend to recommend the paper of Cheng et al (2018) and references therein to readers. We have added some information to make this clear in the revised paper.

Cheng, X. L., Liu, X. M., Liu, Y. J., and Hu, F.: Characteristics of $CO_2$ Concentration and Flux in the Beijing Urban Area, Journal of Geophysical Research: Atmospheres, 123, 1785–1801, 2018.

6) L91-96: since the work of Vickers and Mahrt (1997) is used not only for their quality control method, but also later on as a reference for discussing the results and the approaches to diagnose the non-stationarity, a brief summary of their method and paper content would be worth.

**Response:** We have added a brief summary of their method and paper content in L94-96 in the revised paper.

7) L103-104: the averaging time for this type of analysis is generally chosen as 900, 300 and 60 seconds: the use of a 6-s averaging should be explained and supported; why didn't the authors consider a 60-s-order average? The transition from 900-300 s time averaging to 6s is rather sharp, and it is not surprising that the fluctuation functions differ. It would be interesting to consider an intermediate averaging time.

**Response:** We did not choose the averaging time shown in Fig. 3 according to the principle of equal intervals. The 900s and 300s are chosen because they are commonly used in the eddy-covariance method (for example, see Doran, 2004; Metzger et al, 2007; Li and Bou-Zeid, 2011; Donateo et al, 2017). We choose them to indicate that the IN exists for the commonly used averaging time.

The choice of 6s has two purposes. First, we note that the crossover scale is about 20-30 sec. Thus, one naturally guesses that if using a timescale much smaller than the crossover scale the crossover phenomenon would be changed. In order to show this phenomenon in sharp contrast, we choose a timescale of 6s far away from the crossover timescale and we indeed observed that the crossover moves to a smaller scale. Second, the fact that the inertial-subrange turbulence is also partly removed with the removal of IN can also be shown in sharp contrast (Fig. 3c). This finding clearly reveals the possible relationship between the IN and inertial-subrange turbulence. It can be seen from Fig. 3c that the lowest frequency of inertial subrange is about 0.02Hz-0.03Hz, and the corresponding maximum time scale is about 33s-50s. The 60s scale suggested by the reviewer is not in the inertial subrange, and the choice of 60s cannot achieve above two purposes.

Some discussions on the choice of timescales in Fig. 3 have been added in L105-108 in the revised paper.

Doran, J.C. (2004). Characteristics of Intermittent Turbulent Temperature Fluxes in Stable Conditions. Boundary-Layer Meteorology 112, 241–255.

Metzger M, McKeon B.J and Holmes H (2007) The near-neutral atmospheric surface layer: turbulence and non-stationarity. Phil. Trans. R. Soc. A.365, 859–876.

Li, D., Bou-Zeid, E. (2011). Coherent Structures and the Dissimilarity of Turbulent Transport of Momentum and Scalars in the Unstable Atmospheric Surface Layer. Boundary-Layer Meteorol 140, 243–262.

Donateo, A., Cava, D. and Contini, D (2017). A Case Study of the Performance of Different Detrending Methods in Turbulent-Flux Estimation. Boundary-Layer Meteorol 164, 19–37.

8) L118: an introductory sentence explaining why the OU model is used here would be worth.

**Response:** We used the OU model here for two reasons. First, it has the crossover characteristics. Similar to the data, it is non-stationary at small scales and stationary at large scales. Second, it is very simple. The statistics (including the fluctuation exponents) of this model can be solved analytically (Czechowski and Telesca, 2016). We have added above discussions in L126-128 in the revised paper.

Czechowski, Z. and Telesca, L. (2016). Detrended fluctuation analysis of the Ornstein-Uhlenbeck process: Stationarity versus nonstationarity, Chaos:An Interdisciplinary Journal of Nonlinear Science, 26, 113 109.

9) L130-131: why now a 5-minutes Reynolds average time is used?

**Response:** We here choose 5-min as an example to illustrate the fittings of the OU process and the effect of IN on the test of non-stationarity. The 5-min average time is commonly used in the eddy-covariance method, please see the references listed in the respond of 7). Besides, as we have shown in Fig.3b, as long as the Reynolds average time is not in the inertial subrange, the IN is kept intact in the time series and the results will not change significantly.

---

## Author Response (AR2)

Dear Editors,

We sincerely thank all the Reviewers for reviewing of this paper. The paper has been revised according to the Reviewer comments. We seriously considered these comments and outlined our responds point by point.

We look forward to receiving your opinion.

Yours sincerely,

Lei Liu and Fei Hu

**To Reviewer #1**

1) I don't know if stratification information is available at the tower site but it would be interesting to see how stratification affects the IN of turbulent CO2 fluxes? Or how do the results vary depending on different stratification classes. Turbulent fluxes are largely affected by the stratification. What about IN of turbulent fluxes?

   If no accurate stratification data is available a simple comparison between day and night-time conditions could be a useful extension. During night-time conditions in the atmospheric boundary layers are much more likely to exhibit stable stratification which is a large contrast to the convective day-time boundary layers, particularly in summer.

**Response:** It can be seen from Fig.6 in the paper that the standard deviation of the fluctuation function is small, so we speculate that the IN of time series used in this paper does not seem to be substantially affected by the stratification. According to the Reviewer's suggestion, we plotted the fluctuation functions during day-time and night-time conditions (see Fig. R1). The results indeed show that the IN of time series used in this paper would not be substantially affected by the stability stratification.

   There may be two reasons to explain Fig. R1. Firstly, the IN is found to be intimately related to the small-scale inertial subrange motions of turbulence (see Sec. 3.1 in the paper). According to the cascade theory of Kolmogorov (1941), the

turbulence in the inertial subrange is homogeneous and isotropic, and will not be substantially affected by the anisotropic large-scale motions related to stratification, roughness, and other factors. Cheng et al. (2011) have analyzed the isotropic of turbulence by observations on the Beijing 325-m meteorological tower (data used in our paper is collected on the same tower). They found that the turbulent fluctuations (with scales less than 1 min) are nearly isotropic and large-scale motions (with scales greater than 1 min and less than 10 min) are anisotropic.

[Figure]

**Figure R1. The detrended fluctuation analysis of data during day-time (blue) and night-time (red) conditions.**

Secondly, according to Oke et al. (2017), surface heating caused by urban heat islands and large urban roughness, can generate upward heat fluxes at night, and the nocturnal inversion layer is thus elevated on the top of the constant flux layer (i.e., inertial sublayer) of urban boundary layer. According to Cheng et al. (2018), the inertial sublayer height around the 325-m tower is at least 140 m, and the 80-m height is located in the constant flux layer. Thus, the difference between nighttime and daytime stability stratification is not very obvious at 80 m.

As the Reviewer said, the stability stratification usually affects turbulence. However, the stability stratification is more complex in the megacity than in the rural

areas. We need collected more boundary-layer data to analyze the stratification effect on the IN. This paper mainly focuses on the nonlinear dynamic characteristics of IN, including what the IN is, how to quantify and simulate it by nonlinear dynamic methods, and its impact on flux measurements. In another paper, we will discuss the boundary-layer meteorological characteristics of IN in details, including the impact of stratification, roughness, and other boundary-layer parameters on IN, when more boundary-layer data are collected and analyzed.

**Reference:**

Kolmogorov, A. N. (1941), The local structure of turbulence in incompressible viscous fluid for very large Reynolds numbers, Dokl. Akad. Nauk SSSR, 30, 301–305.

Cheng, X. L., Q.-C. Zeng, and F. Hu (2011), Characteristics of gusty wind disturbances and turbulent fluctuations in windy atmospheric boundary layer behind cold fronts, J. Geophys. Res., 116, D06101.

Oke T. R., Mills G., Christen A., and Voogt J. A. (2017) Urban Climate, Cambridge University Press.

Cheng, X. L., Liu, X. M., Liu, Y. J., and Hu, F. (2018) Characteristics of CO2 concentration and flux in the Beijing urban area, J. Geophys. Res., 123, 1785–1801.

**2)** Generally, and also due to the 2nd comment above, the manuscript would benefit from a short description of the turbulent and typical boundary layer structures of the Beijing downtown urban boundary layer. Is the turbulence usually isotropic? are stable conditions common? etc...

**Response:** We have added some descriptions of turbulence and typical boundary layer structures of the Beijing downtown. These descriptions are based on the long-term observations of the 325-m tower and have been mentioned in the Response of 1). As the Reviewer said, the added information can indeed help readers better understand the content of this paper. Please see the second paragraph in Sec. 2.3 of the revised paper.

**3)** Technical comments

**Response:** We have revised the mistakes or ambiguous statements according to your suggestions. Thank you!